# Microstructure Evolution, Constitutive Modelling, and Superplastic Forming of Experimental 6XXX-Type Alloys Processed with Different Thermomechanical Treatments

**DOI:** 10.3390/ma16010445

**Published:** 2023-01-03

**Authors:** Andrey G. Mochugovskiy, Ahmed O. Mosleh, Anton D. Kotov, Andrey V. Khokhlov, Ludmila Yu. Kaplanskaya, Anastasia V. Mikhaylovskaya

**Affiliations:** 1Physical Metallurgy of Non-Ferrous Metals, National University of Science and Technology “MISIS,” Leninsky Prospekt, 4, 119049 Moscow, Russia; 2Mechanical Engineering Department, Faculty of Engineering at Shoubra, Benha University, Cairo 11629, Egypt; 3Institute of Mechanics, Lomonosov Moscow State University, Michurinskiy Prospect, 1, 119192 Moscow, Russia

**Keywords:** microstructure, superplasticity, mathematical modeling, cavitation

## Abstract

This study focused on the microstructural analysis, superplasticity, modeling of superplastic deformation behavior, and superplastic forming tests of the Al-Mg-Si-Cu-based alloy modified with Fe, Ni, Sc, and Zr. The effect of the thermomechanical treatment with various proportions of hot/cold rolling degrees on the secondary particle distribution and deformation behavior was studied. The increase in hot rolling degree increased the homogeneity of the particle distribution in the aluminum-based solid solution that improved superplastic properties, providing an elongation of ~470–500% at increased strain rates of (0.5–1) × 10^−2^ s^−1^. A constitutive model based on Arrhenius and Beckofen equations was used to describe and predict the superplastic flow behavior of the alloy studied. Model complex-shaped parts were processed by superplastic forming at two strain rates. The proposed strain rate of 1 × 10^−2^ s^−1^ provided a low thickness variation and a high quality of the experimental parts. The residual cavitation after superplastic forming was also large at the low strain rate of 2 × 10^−3^ s^−1^ and significantly smaller at 1 × 10^−2^ s^−1^. Coarse Al_9_FeNi particles did not stimulate the cavitation process and were effective to provide the superplasticity of alloys studied at high strain rates, whereas cavities were predominately observed near coarse Mg_2_Si particles, which act as nucleation places for cavities during superplastic deformation and forming.

## 1. Introduction

The 6000-type Al-Mg-Si-Cu-based alloys are extensively used in the transportation industry due to their high specific strength [1,2]. The alloys belong to a group of heat-treatable aluminum-based alloys strengthened by metastable β´(Mg_2_Si) and Q´(AlMgSiCu) phases during T6 treatment, including solid solution treatment and aging [3,4,5,6]. The precipitation strengthening effect gives the 6000-type alloys a considerable advantage over 5000-type and 3000-type alloys. These alloys are attractive for the superplastic forming of the complex-shaped parts due to a low critical cooling rate providing the supersaturated solid solution.

In the 6000 alloys, the sum of the concentrations of the key alloying elements of Mg, Si, and Cu is usually below 3 wt.%. The low solute contributes to the corrosion resistance and processing properties. However, alloys with a low solute content have a tendency to intense grain growth that weakens the elevated temperature formability. For high-strain rate superplastic forming and quick plastic forming techniques, fine-grained and ultrafine-grained alloys are required [7,8]. The optimization of both chemical composition and thermomechanical treatment helps to form a fine-grained and thermally stable structure [9,10,11,12,13]. The approach to designing fine-grained and ultrafine-grained alloys has been repeatedly reported. Alloying with transition metals (TM), including rare earth (RE) elements, led to a grain refinement through the combination of particle-stimulated nucleation (PSN) [14,15,16] and Zener pinning [17,18,19] mechanisms. The coarse particles of ~1 µm in size provide a PSN effect. The coarse particles are formed due to the fragmentation of eutectic-originated phases in thermomechanical treatment. The fine dispersoids of 5–50 nm in size that precipitate during decomposition of the supersaturated by TM/RE solid solution in a thermomechanical treatment [20,21,22,23,24,25,26] lead to a Zener pinning effect. Dispersoids pin grain boundaries and inhibit grain growth. This approach is realized for 5000-type [27,28,29,30], 6000-type [31,32,33,34], 7000-type [35,36,37], and 2000-type [38,39] alloys. Thermomechanical treatment should provide a uniform distribution of the coarse particles in the aluminum matrix and a high number density of nanoscale-sized dispersoids. Thus, the thermomechanical treatment has an important role in microstructure evolution and final grain refinement. An intermediate heterogenization annealing and a high degree of cold/warm rolling are required to realize the PSN effect and to form a fine-grained structure in the conventional high-alloyed Al-Zn-Mg-Cu based 7075 alloy [14], Al-Mg-based alloys. Oppositely, an experimental Al-Zn-Mg-Cu-based alloy with Ni and a high fraction of coarse Al_3_Ni particles demonstrated a fine-grained structure and good superplastic properties after both small (10–20%) and large (70%) cold rolling degrees [36]. A recently developed Al-Mg-Si-Cu-based alloy alloyed with Fe and Ni, and minor Sc and Zr additions, exhibited high strength and demonstrated a fine-grained structure and superplasticity in a strain rate range of 2 × 10^−3^–2 × 10^−2^ s^−1^ and a temperature range of 440–520 °C [32]. A high number density of nanoscale precipitates of the Al_3_(Sc,Zr) phase provided a strong Zener pinning effect, and they are formed during the low-temperature annealing of as-cast alloy [40]. Owing to low temperature annealing, the residual non-equilibrium Mg_2_Si phase and Al_3_(Sc,Zr) nanoscale precipitates worsen ductility at room temperature and limit the processing properties of the alloy. A decreased cold rolling degree can help to overcome the processing problems, but cold rolling can be a principal operation that is required for the accumulation of high-store energy of recrystallization during further superplastic flow and fine-grained structure formation [15]. Therefore, it is necessary to study the influence of the hot/cold rolling degree ratio on the grain structure and superplastic properties of the novel Al-Mg-Si-Cu-based alloy.

The characterization of the stress-strain behavior involves a mathematical description of the superplastic deformation process that is important for a successful forming operation. The mathematical modeling helps to simulate superplastic forming based on the finite element method and to develop forming regimes for complex-shaped parts. A model of the deformation behavior provides a “bridge” between material properties and forming processes to make high-quality complex-shaped metallic components with uniform thickness distribution. The models based on the Zener–Hollomon parameter and Arrhenius-type equations are widely utilized to describe the hot deformation behavior of materials characterized by a near stable flow [41,42,43,44].

The purposes of the current study included (1) the study of the effect of hot/cold rolling degree on the grain structure and superplasticity of the novel Al-Mg-Si-Cu-Fe-Ni-Zr-Sc alloy to choose appropriate treatment and conform the high strain rate formability of the alloy, and (2) to describe the deformation behavior of the alloy with a constitutive mathematical model.

## 2. Materials and Methods

The alloy of the following composition of Al-1.2 wt.%Mg-0.7 wt.%Si-0.9 wt.%Cu-1.0 wt.%Fe-1.0 wt.%Ni-0.2 wt.%Zr-0.1 wt.%Sc was prepared in a laboratory inductive furnace (Interselt, Saint-Petersburg, Russia) using a graphite-fireclay crucible (Lugaabrasiv, Luga, Russia). The casting was processed in a water-cooling copper mold with an internal size of 100 × 40 × 20 mm. The melt was prepared using the following pure metals: 99.99 wt.%Al, 99.95 wt.%Mg, and master alloys of Al-20 wt.%Ni, Al-10 wt.%Fe, Al-12 wt.%Si, Al-2 wt.%Sc, Al-5 wt.%Zr, and Al-53.6 wt.%Cu. Before casting, the melt was heated to 800 °C. The temperature during solidification was controlled using a chromel–alumel thermocouple, and the cooling rate was ~15 K/s.

The obtained ingots were subjected to one- or two-stage heat treatment. The first stage was performed at 350 °C for 8 h, and the second stage was carried out at 480 °C for 3 h. The annealed samples were thermomechanically treated with 4 various regimes, including hot rolling (Rolling mill V-3P, GMT, Saint-Petersburg, Russia) at 450 °C (HR) and cold rolling (CR) at room temperature in different proportions (Table 1). The final thickness of sheets was 1.00 ± 0.05 mm.

The samples for microstructural examination were prepared using a Struers LaboPoll-5 polishing machine via mechanical grinding on SiC papers (grit sizes of 320, 800, 1200, 2400, 4000) and final polishing with an OP-S silica-based colloidal suspension (grain size of 0.04 µm). Scanning electron microscopy (SEM) in a Tescan-VEGA3 LMH (Tescan Brno s.r.o., Kohoutovice, Czech Republic) and light optical microscopy (OM) in a Zeiss Axiovert 200 M (Carl Zeiss, Oberkochen, Germany) were used for the microstructural examination. SEM was equipped with an energy dispersive X-ray spectrometer (EDS) X-MAX80 (Oxford Instruments plc, Abingdon, UK) and an EBSD-detector HKL NordlysMax (Oxford Instruments plc, Abingdon, UK). The grain structure was studied with OM in a polarized light. For this purpose, the pre-polished samples were anodized at a voltage of 18 V in a Barker’s solution for 60 s at a temperature of 2 °C below zero. The EBSD maps were generated from an area of 150 × 150 μm^2^ using a step size of 0.3 μm. The samples for microstructural examination were prepared using a Struers La-boPoll-5 polishing machine via mechanical grinding on SiC papers and polishing in an OP-S silica-based colloidal suspension. The transmission electron microscopy (TEM) was performed using a JEOL JEM 2100 microscope (JEOL, Tokyo, Japan). For the TEM analysis, disc-type samples that were 3 mm in diameter and 0.22 ± 0.01 mm thick were used. The samples were electrochemically thinned in a methanol solution of 30% nitric acid using a Struers TenuPol-5 twinjet machine (Struers APS, Ballerup, Denmark) at a temperature of minus 20 ± 1 °C and a voltage of 19 ± 2 V.

The mean particle/dispersoid size (equivalent diameter), volume fraction, and aspect ratio were calculated using AxioVision Vs.40 V4.5.0.0 software (Carl Zeiss, Oberkochen, Germany). The interparticle space was calculated using a linearly secant method in a longitudinal cross section perpendicularly to the rolling direction. At least 40 secants were used for each state. The channel 5 software (Oxford Instruments plc, Abingdon, UK) was used to calculate the mean grain size for the EBSD data. To determine the size of the L1_2_ phase precipitates, high-resolution TEM images and dark field images were used. The number of measurements was in the range of 200–900 for grains and subgrains, 300–400 for eutectic originated particles, and 400–500 for dispersoids. The error bars for a mean value were calculated as a confidence interval with a confidence probability of 0.95. 

The superplastic properties of the studied material were analyzed using a Walter Bai LFM-100 machine (Walter + Bai AG, Löhningen, Switzerland) through a uniaxial tensile test with a constant strain rate and periodically stepped strain rate following the ASTM-E2448–11 standard. The tensile test with a constant strain rate was performed in a strain rate range of 2 × 10^−3^–1 × 10^−2^ s^−1^. The strain rate was maintained constant by an increasing crosshead velocity that was proportional with an increase in the length of the gage part of the sample. To identify the strain rate sensitivity *m*-coefficient and its strain-induced evolution, the step tests, in which strain rate was periodically stepped to 20% above nominal and then back to nominal every 0.1 strain. The sample gage part width was 6 mm, the thickness was 1 mm, and the length was 14 mm. Three samples per point were tested. The stress–strain curves were used to construct a constitutive model to predict the superplastic flow behavior of the alloy. To evaluate the quality of the model, a correlation coefficient R2, an average absolute relative error (AARE), and a root mean square error (RMSE) were calculated [45,46,47]. Higher correlation coefficients and reduced error are noteworthy indicators of the quality of the models. The results of the model and experiments were fed into DEFORM 3D v.6.1 software (Scientific Forming Technologies Corporation, Columbus, USA) to simulate the forming process and determine a pressure-time regime. Superplastic forming was processed at 480 °C, and strain rates of 2 × 10^−3^ and 1 × 10^−2^ s^−1^ were processed in a laboratory forming machine with control of the Ar gas pressure and temperature of the process using the regime obtained by DEFORM 3D based on the data of the tensile tests. The shape and dimensions of the used mold were described in our previous works [48,49]. The used mold shape was designed to have a critical region with different strain rates to evaluate the superplasticity of this region and to assess the thickness difference. 

## 3. Results

### 3.1. Microstructure of the Alloy

The as-homogenized structure of the studied alloy is shown in Figure 1. After the first homogenization stage at 350 °C, the aluminum-based solid solution (Al) and the Al_9_FeNi and Mg_2_Si phases were observed. The volume fractions of the Al_9_FeNi and Mg_2_Si phases were 4.5 ± 0.5% and 2.0 ± 0.2%, respectively. Sc and Zr were dissolved in the Al-based solid solution, and the solidification origin phases enriched with these elements were not found. After the second step of annealing at 480 °C, the volume fractions of the Al_9_FeNi and Mg_2_Si phases were 4.4 ± 0.5% and 1.8 ± 0.2%, respectively. The second annealing stage led to the fragmentation and spheroidization of the Al_9_FeNi and Mg_2_Si particles and the partial dissolution of the Mg_2_Si phase. The microstructures of the studied alloy with 0.1 wt.%Sc in as-cast and as-annealed states were similar to the alloy with a higher Sc content of 0.2 wt.% [32]. 

The TEM study of the samples annealed at 350 °C for 8 h revealed a high density of the nanoscale precipitates (Figure 2a,b). The mean size of precipitates was 10 ± 1 nm. The selected area electron diffraction (SAED) (Figure 2c) exhibiting ordered superlattice reflections confirmed the L1_2_ structure for precipitates. The Al [011] zone axis was parallel to the [011] zone axis of the L1_2_ phase. The second annealing step at 480 °C for 3 h increased the mean size of precipitates to 13 ± 1 nm (Figure 2) but did not influence their structural type, which was confirmed by SAED (Figure 2f) and Fast Fourier Transform patterns (Figure 2i).

After thermomechanical treatment, the eutectic-originated phases were fragmentized independently on the treatment regime. The parameters of particles for various regimes are shown in Table 2. As a result, particles with a size in the range of 1.1–1.7 µm surrounded by the aluminum solid solution were formed (Figure 3). The mean size of the Al_9_FeNi particles was 0.9 ± 0.1 µm for hot rolled samples (1HR and 2HR) independently on the homogenization regime, and it was 0.7 ± 0.1 µm for cold rolled samples (CR50 and CR80); the particle aspect ratio was 0.80 for hot rolled samples (1HR and 2HR), and lower values of 0.72 were observed for cold rolled samples (CR50 and CR80). For the Mg_2_Si phase, the mean particle size was 0.8 ± 0.1 µm, and the particle aspect ratio was 0.8 for the studied treatment regimes, including two-step homogenization, and finer particles of 0.5 ± 0.1 µm with a lower aspect ratio of 0.71 were observed in the samples pre-homogenized at a low temperature in one step. The mean values of the interparticle spaces were 1.4 ± 0.3, 1.2 ± 0.1, 1.1 ± 0.2, and 1.1 ± 0.3 µm with a standard deviation of 0.9, 0.5, 0.7, and 0.9 µm for the 1HR, 2HR, CR50, and CR80 regimes, respectively. The treatment regime 2HR including two-step annealing and hot rolling with 90% reduction, providing particles of about 1 µm and the most homogeneous microstructure with a low deviation of the interparticle space compared to the other treatment regimes (Figure 3a,b). 

### 3.2. Superplastic Deformation Behavior 

The true stress vs. true strain dependencies obtained by the constant strain rates in a range of 2 × 10^−3^–1×10^−2^ s^−1^ and at a temperature of 480 °C are shown in Figure 4. The m > 0.3 was observed for all studied regimes in the studied strain range. The hot rolled samples (1HR and 2HR regimes, Figure 4a,b) provided a stable flow behavior during the test, whereas strain hardening was observed for the samples processed with cold rolling (CR50 and CR80 regimes, Figure 4c,d). 

The m-value varied within 0.33–0.45 for the HR1 regime (Figure 4a), 0.37–0.44 for HR2 (Figure 4b), 0.32–0.45 for R50 (Figure 4c), and 0.29–0.41 for R80 (Figure 4d). Herewith, the 2HR regime provided a stable m-value during the test, whereas for CR50 and CR80, the m value decreased with the strain increase. 

The elongation-to-failure values for the samples processed with different treatments are shown in Table 3. The elongations reached from 350 to 470%, and the maximum value was observed at a strain rate of 1 × 10^−2^ s^−1^ for all treatment regimes. A more uniform and stable flow with large elongations even at a high strain rate of 2 × 10^−2^ s^−1^ were observed for the samples treated with the 2HR regime. 

For the 2HR-treated samples, which demonstrated a good superplasticity, the stress–strain behavior was studied in the wider temperature and strain rate ranges of 440–500 °C and 2 × 10^−3^–2 × 10^−2^ s^−1^ (Figure 5). The increase in temperature and decrease in strain rate resulted in a decrease of flow stress values. A larger elongation with stable flow was revealed at the temperatures of 460–480 °C compared to 440 and 500 °C (Table 4). 

### 3.3. Constitutive Modeling of the Superplastic Deformation 

Modeling the flow stress behavior during the deformation helps to reduce time, efforts, trials, materials, and the manufacturing cost. To describe the strain rate ($\dot{\varepsilon }$) dependence vs. stress (*σ*) during the deformation that occurs at limited temperatures and strain rates and that is characterized by a small stress, Equation (1) [50,51] was used.
(1)$$\dot{\varepsilon }=A\sigma^{n}\exp^{\left(-\frac{Q}{RT}\right)}$$
where *A* and *n* are the material constants depending on strain; *Q* (J/mol) represents the apparent activation energy and also depends on strain; *R* is universal gas constant is 8.314 J/(mol K), $\dot{\varepsilon }$ in s^−1^, *σ* in MPa, and *T* in K.

The experimental stress–strain curves were divided in two groups: group A was used in constructing the model and calculating the equation constants, and group B was used to assess the predictability of the constructed model. For reliable checking, the second part was selected to include different temperatures and different strain rates, including four testing conditions: 0.005 s^−1^ and 440 °C; 0.008 s^−1^ and 460 °C; 0.01 s^−1^ and 480 °C; and 0.002 s^−1^ and 500 °C. Equation (2) was created, as follows (Equation (3)), by taking the natural logarithm of both sides:(2)$$\ln(\dot{\varepsilon })=\ln\left(A\right)+n\ln\left(\sigma \right)-\left(\frac{Q}{R}\left(T^{-1}\right)\right)$$

For determining *n*, the partial differentiation of $\ln(\dot{\varepsilon })$ with respect to ln(σ) should be used. At a constant temperature, Equation (3) can be expressed as follows: (3)$$n={\left[\frac{\partial \ln\dot{\varepsilon }}{\partial \ln\sigma }\right]}_{T=const}$$

For determining *Q*, the partial differentiation of Equation (2) with respect to $T^{-1}$ should be used, and Equation (3) is expressed by (Equation (4)): (4)$$\frac{\partial \ln(\dot{\varepsilon })}{\partial \ln\left(T^{-1}\right)}=\frac{\partial \ln\left(A\right)}{\partial \ln\left(T^{-1}\right)}+n\left(\frac{\;\partial \ln\left(\sigma \right)}{\partial \ln\left(T^{-1}\right)}\right)-\left(\frac{Q}{R}\frac{\;\partial T^{-1}}{\partial \ln\left(T^{-1}\right)}\right)$$

For a constant strain rate: (5)$$0=0+n\;\frac{\partial \ln\left(\sigma \right)}{\partial \ln\left(T^{-1}\right)}-\frac{Q}{R}\Rightarrow \;\frac{Q}{R}=n\;\frac{\partial \ln\left(\sigma \right)}{\partial \ln\left(T^{-1}\right)}$$
(6)$$Q=R\times {\left[\frac{\partial \ln\dot{\varepsilon }}{\partial \ln\sigma }\right]}_{T}\times {\left[\frac{\partial \ln\sigma }{\partial \left(T^{-1}\right)}\right]}_{\dot{\varepsilon }}$$

Finally, the values of the flow stress were calculated as follows, according to the simple power law (Equation (7)):(7)$$\sigma ={\left(\frac{z}{A}\right)}^{\frac{1}{n}}$$
where $Z=\dot{\varepsilon }\times \exp\left(\frac{Q}{RT}\right)$ is the Zener-Holomon parameter. The strain dependencies of the material constants of *Q*, *n*, and *A* are illustrated in Appendix A, and the corresponding polynomial fitting parameters in Equation (8) are presented in Appendix A. The fifth polynomial equation provided a good fitting with the lower error (R2 = 0.98–1.0) compared to the third and fourth polynomial equations.
(8)$$\{\begin{array}{c} Q=Y_{10}+B_{11}\varepsilon^{1}+B_{12}\varepsilon^{2}+B_{13}\varepsilon^{3}+B_{14}\varepsilon^{4}+B_{15}\varepsilon^{5}\\ n=Y_{20}+B_{21}\varepsilon^{1}+B_{22}\varepsilon^{2}+B_{23}\varepsilon^{3}+B_{24}\varepsilon^{4}+B_{25}\varepsilon^{5}\\ A=Y_{30}+B_{31}\varepsilon^{1}+B_{32}\varepsilon^{2}+B_{33}\varepsilon^{3}+B_{34}\varepsilon^{4}+B_{35}\varepsilon^{5} \end{array}$$

Figure 6 shows the stress-strain curves obtained from experiments and the model. The presented curves revealed a high approximation accuracy of the model in fitting the data of group A, which was used for this model. The validation of the model proved the high predictability of the unmodeled data of group B (Figure 7a). The statistical comparisons between the experimental data and the fitted (group A) and expected (group B) data confirm the excellent capability of the constructed simple law model in fitting and prediction (Figure 7b,c). The comparison indices, R2, AARE (%), and RMSE, after fitting group A and prediction group B are 0.98, 1%, 0.5 and 0.97, 4%, 0.7, respectively. 

The constructed model was used in predicting the untested data, and model data were fed into a finite element simulator (Deform 3D) to adjust the material performance inside the simulator for the successful simulation of the superplastic forming process (see Section 3.5).

### 3.4. The Microstructural Evolution during Superplastic Deformation

To analyze the microstructure before the start of the superplastic deformation, the thermomechanically treated sheets were annealed at 480 °C for 20 min, followed by cooling with cold water. The samples exhibited partly recrystallized grain structure at elevated temperatures (Appendix A), which was the result of the Zener pinning effect of nanoscale dispersoids. 

The EBSD grain boundary maps and misorientation angle distributions after 200% of superplastic deformation at 480 °C and a strain rate of 1 × 10^−2^ s^−1^ demonstrated an almost recrystallized structure with a large fraction of high-angle grain boundaries (Figure 8). Thus, dynamic recrystallization occurred during the superplastic deformation. For the 1HR, 2HR, CR50, and CR80 regimes, the HAGB fractions were 80–85%, the mean subgrain size was 3 ± 1 µm, and the mean grain size was 4 ± 1 µm for all studied samples. There were no significant differences between grain sizes and misorientation angle distributions for the deformed samples treated with different modes. 

The fraction of cavities after the 200% deformation (ε = 1.1) was 1.5 ± 0.8% for hot rolled samples (1HR and 2HR) and 1.9 ± 0.6% for the samples processed with cold rolling (CR50 and CR80). Cavities were predominantly formed near the Mg_2_Si particles (Figure 9). 

### 3.5. Superplastic Forming

All stress–strain results, experimental data, and predicted data for the untested conditions from the constructed model were fed into the DEFORM-3D software to define and adjust the material characteristics for finite element simulation and to determine the forming regimes and pressure-time dependence corresponding to 1 × 10^−2^ s^−1^ and 2 × 10^−3^ s^−1^ and strain distributions after the process. 

The superplastic forming (SPF) of the thin-walled complex-shaped part was performed using sheets processed with the 2HR regime. Geometry for the SPF part was chosen according to [45]. The presented geometry exhibited a complex shape with a high strain, and metallic parts of a such shape are difficult to process with traditional forming methods. The median cross-section of the parts and thickness distributions are presented in Figure 10. Importantly, the thickness distribution for the high strain rate SPF at 1 × 10^−2^ s^−1^ was more uniform than that for the low strain rate SPF at 2 × 10^−3^ s^−1^. 

The experimental FSP and FES results are identical with small differences (Figure 10), and the errors do not exceed 10%. Therefore, the constructed model is recommended to be used for predicting the flow behavior of this alloy without performing experiments, and these data can be fed into any FE simulator for process simulation. Thus, the right modeling flow behavior could decrease the time, raw materials, energy, and the manufacturing cost.

The microstructural analysis revealed that forming at a strain rate of 2 × 10^−3^ s^−1^ led to a higher residual cavitation of 8.5% compared to 2.1% at a strain rate of 1 × 10^−2^ s^−1^ (Figure 11). It should be noted that the cavities were observed near the Mg_2_Si particles, while Al_9_FeNi particles did not initiate cavitation, similarly to tensile tests. 

## 4. Discussion

Grain growth and dynamic grain growth were pronounced for Al-Mg-Si-based alloys [52,53]. Due to low solute content at a superplastic deformation temperature and the presence of high diffusive Si atoms, a high number density of nanoscale precipitates was a critical component that provided the superplasticity of the studied alloys [31,54,55]. Alloying with Sc and Zr is an effective combination of the alloying elements to form nanoscale L1_2_ precipitates for a strong Zener pinning effect. Considering data of [56,57], the Si atoms may substitute Al in the L1_2_ phase to form the (Al,Si)_3_(Sc,Zr) phase. It is notable that Si insignificantly influences the mean size of precipitates for the studied alloys. A similar mean size of the Al_3_(Sc,Zr) precipitates was observed after annealing at 350 °C in Si-free Al-Mg-Zr-Sc-based alloys [40]. A similar L1_2_ phase precipitate size was observed for the Al-Mg-Si-Cu-Fe-Ni-Zr-Sc alloy with 0.2%Sc [32]. Therefore, the 0.7%Si and change in the Sc in a range of 0.1–0.2 wt.% insignificantly influenced the precipitate size. Finer precipitates of the L1_2_ phase in the studied alloy than that of in Al-Si-Sc alloys [57,58] can be explained by the Zr core. Zr atoms, due to a low diffusion rate in Al, stabilize the size of L1_2_ structure [59]. The alloys exhibited a similar size to L1_2_ precipitates after both the one-step and two-step homogenization of 10–13 nm. Due to L1_2_ precipitates, the microstructure of the alloy studied was almost non-recrystallized independently on the treatment regime, and dynamic recrystallization occurred during the superplastic deformation. 

Coarse particles of both Al_9_FeNi and Mg_2_Si are also important microstructural components, proving superplasticity at high strain rates for the studied alloy. The role of the coarse particles in the superplastic deformation behavior of the aluminum-based alloys with initial non-recrystallized grain structure is discussed in [14]. Coarse particles led to the PSN effect during superplastic deformation and stimulated dynamic recrystallization. Thus, coarse particles provide fine equiaxed recrystallized grains that are required for successful grain boundary sliding and superplastic behavior [60]. The thermomechanical treatment had a noticeable influence on the particle distribution and superplasticity of the alloy. The homogeneous distribution of the particles is important for homogeneous recrystallized grain structure and stable superplastic flow at high strain rates. First, the two-step homogenization provided the fragmentation and spheroidization of the coarse particles. Second, the increase in the hot/cold rolling ratio increased the particle distribution homogeneity and favored a uniform particle distribution after rolling. Thermomechanical treatment with cold rolling did not provide an advantage compared to only hot rolling. High particle distribution homogeneity resulted in a uniform grain structure during superplastic deformation and better superplasticity. 

The additional parameter that controlled the superplastic behavior was cavitation. The microstructural study of the samples subjected to the superplastic deformation and superplastic forming revealed the predominate nucleation of cavities on the particles of the Mg_2_Si phase. The samples processed with low temperature homogenization and with a higher fraction and a lower aspect ratio of the Mg_2_Si particles demonstrated a higher residual cavitation and weaker superplasticity. Thus, appropriate treatment regimes should decrease the fraction of the Mg_2_Si phase for the better superplasticity of aluminum-based alloys. Cavities were not observed near similarly coarse particles of the Al_9_FeNi phase; therefore, this phase helped with grain refinement, did not initiate cavitation, and improved superplasticity. It is well known that Mg_2_Si initiates cracking growth and decreases the processing properties and ductility of Al-based alloys [61,62]. Alloys with a eutectic-originated Al_9_FeNi phase, including industrial AA2618 [63,64], are successfully processed with thermomechanical treatments and demonstrate good ductility and cracking resistance [27,65]. The difference in the particle effect can be explained by the difference in the interphase energy of (Al)/Mg_2_Si and (Al)/Al_9_FeNi couples.

The components were successfully processed by SPF from the studied alloy with both low and high strain rates without failure. For the presented sample geometry after the superplastic forming, large strains were realized between points 4 and 5 and between points 6 and 7. In these critical areas, the thickness distribution was usually less homogeneous [45]. A significant difference in the fraction of the residual cavitation was observed in the formed parts. The increase in strain rate from 2 × 10^−3^ to 1 × 10^−2^ s^−1^ reduced the cavitation significantly, from 8.5 to 2.1%. Lower cavitation was a reason for stable flow and a more uniform thickness distribution in the critical zones for the high strain rate. The effect can be explained by finer grains that formed during the superplastic deformation at higher strain rates for the alloys with initially non-recrystallized grain structure in the studied alloy [32]. 

There are many studies that developed hyperbolic sine-typed Arrhenius models [46,66,67,68,69,70,71,72,73,74] and Johnson-Cook type models [75,76,77] for the successful prediction of the hot deformation behavior of different materials, including superplastic Al and Ti alloys. The simple power law function, with a smaller number of constants, was successfully used for predicting the flow behavior of Ti-based alloys [66]. For comparatively narrow temperature–strain rate ranges of superplastic conditions, the developed model that was based on the simple power equation (Beckofen) also demonstrated a low error level. The disadvantage of the model is the application of the polynomial function with many coefficients, and the effective activation energy (Q) and strain rate exponent (n) demonstrated a complicated dependency from strain with the maximum strain value of about 0.4 due to significant changes in the microstructure, with the cooperation of dynamic recrystallization and dynamic grain growth. Further efforts should focus on considering particular microstructural parameters and their strain-induced evolution during modeling process, which are required to improve model predictability and effectiveness.

## 5. Conclusions

The increase in proportion of hot/cold rolling degrees increased the uniformity of the distribution of the eutectic-originated particles of the Al_9_FeNi and Mg_2_Si phases and improved the superplastic properties of the studied Al-Mg-Si-Fe-Ni-Zr-Sc alloy. Homogenization at 480 °C and a hot rolling reduction of 70–90% were required to form the homogeneous distribution of the eutectic-originated particles. Due to a high number density of nanoscale-sized Al_3_(Sc,Zr) and coarse eutectic-originated particles, a fine grain structure of the studied alloys formed during superplastic deformation, and a high degree of cold rolling, such as that used for many superplastic alloys, was not required.

The superplastic properties were studied, and the deformation behavior was described by the Arrhenius model based on the power equation of the stress dependences vs. strain rate (Beckofen equation) for the alloy studied. The maximum elongation-to-failure of ~470–500% was observed at (0.5 − 1) × 10^−2^ s^−1^ and at a temperature of 460–480 °C for the samples processed with a high hot rolling reduction.

The superplastic forming of the complex-shaped thin-wall components was successfully modeled and processed at low 2 × 10^−3^ and high 1 × 10^−2^ s^−1^ strain rates. The modeling and experimental results demonstrated better formability of the alloy studied at a higher strain rate, with 1.5-times lower residual cavitation and 1.3-times higher uniformity of the thickness distribution. Cavities were observed near particles of the eutectic-originated Mg_2_Si phase, whereas Al_9_FeNi phase particles did not initiate cavitsation.

## Figures and Tables

**Figure 1 materials-16-00445-f001:**
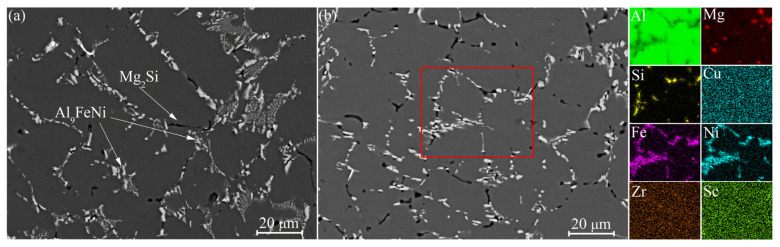
SEM images (backscattered electrons) of the samples annealed at (**a**) 350 °C for 8 h and (**b**) 350 °C for 8 h and a second step at 480 °C for 3 h.

**Figure 2 materials-16-00445-f002:**
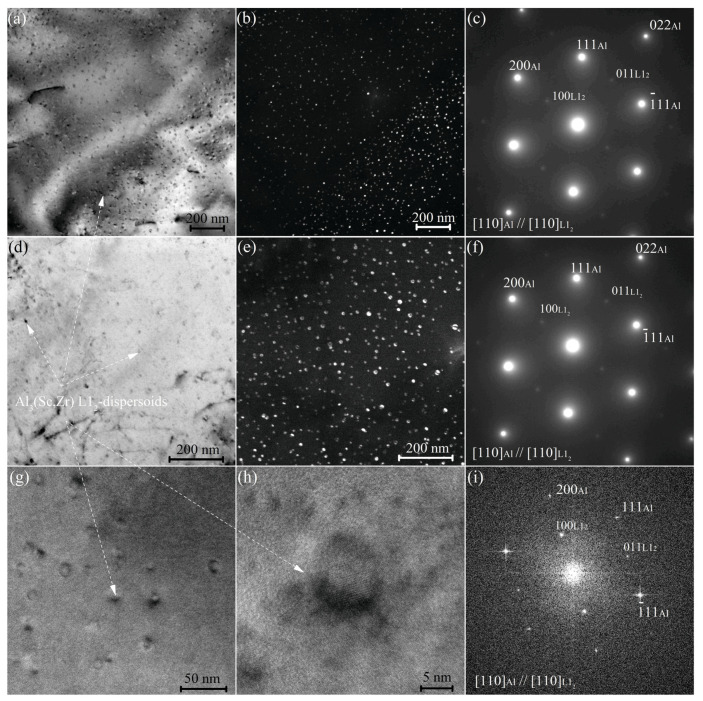
TEM images for the samples (**a**–**c**) after one-stage annealing at 350 °C for 8 h and (**d**–**i**) for two-step annealing with the first stage at 350 °C for 8 h and the subsequent second stage at 480 °C for 3 h; (**a**,**d**,**g**) bright fields, (**b**,**e**) dark fields, (**c**,**f**) SAEDs, (**h**,**i**) high resolution with corresponding FFT.

**Figure 3 materials-16-00445-f003:**
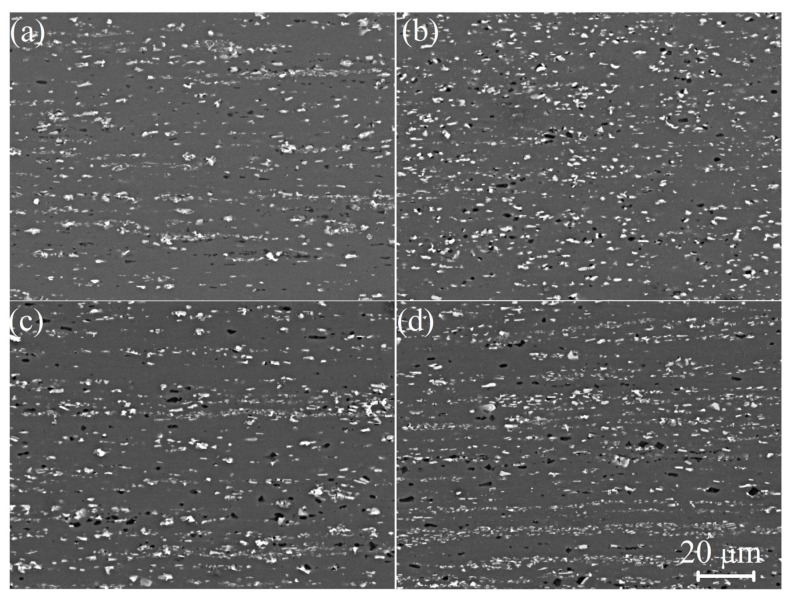
SEM images of the samples thermomechanically treated in (**a**) 1HR, (**b**) 2HR, (**c**) CR50, and (**d**) CR80 regimes.

**Figure 4 materials-16-00445-f004:**
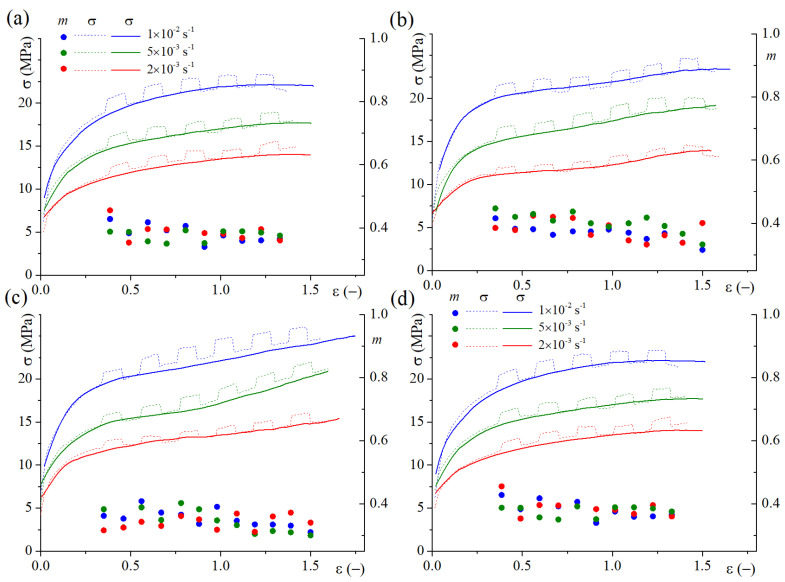
Dependencies of stress and m-value on the strain obtained with a constant strain rate test and a test with a periodically stepped strain rate 20% above the nominal of 2 × 10^−3^ s^−1^, 5 × 10^−3^ s^−1^, and 1 × 10^−2^ s^−1^ for (**a**) 1HR, (**b**) 2HR, (**c**) CR50, and (**d**) CR80 regimes for a temperature of 480 °C.

**Figure 5 materials-16-00445-f005:**
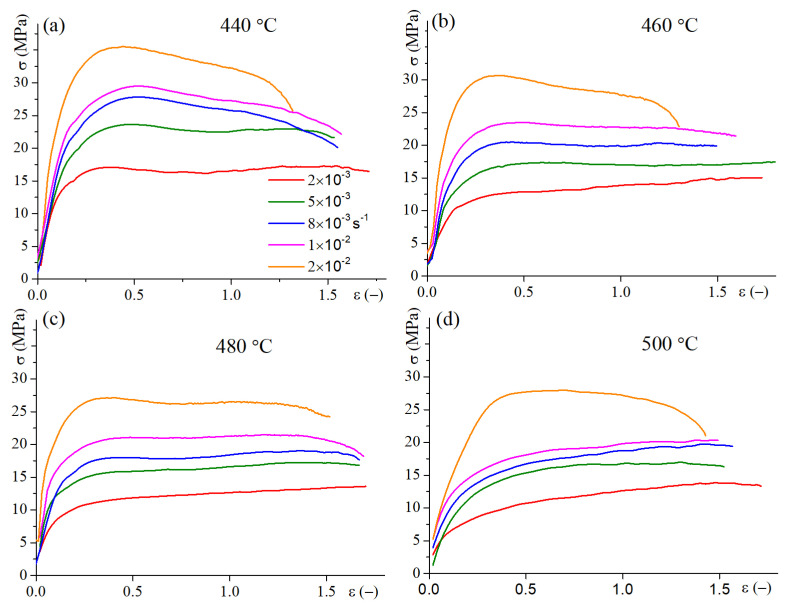
Stress-strain dependencies with constant strain rates in a range of 2 × 10^−3^ to 2 × 10^−2^ s^−1^ and temperatures of (**a**) 440 °C; (**b**) 460 °C; (**c**) 480 °C; (**d**) 500 °C for samples processed with the 2HR regime.

**Figure 6 materials-16-00445-f006:**
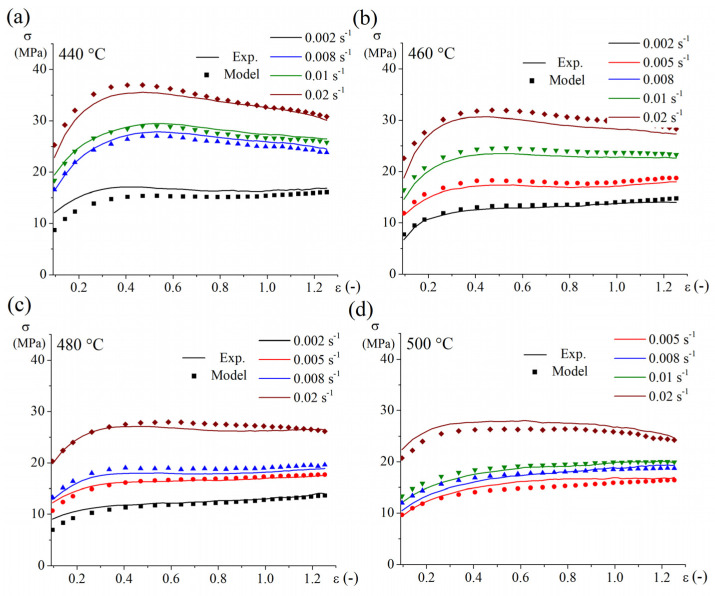
The stress-strain dependency of the experimental data and approximated data by Equation (8) (group A) for different strain rates at temperatures of (**a**) 440, (**b**) 460, (**c**) 480, and (**d**) 500 °C.

**Figure 7 materials-16-00445-f007:**
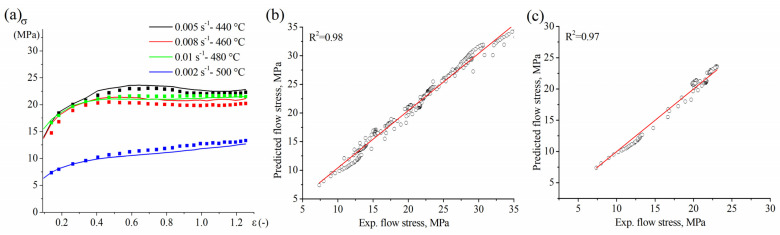
(**a**) The stress-strain dependency of the experimental data and predicted data by Equation (8) of the unmodeled data (group B), (**b**) the correlation between experimental data and the fitted flow stress of group A, and (**c**) the correlation between experimental data and expected the flow stress of group B.

**Figure 8 materials-16-00445-f008:**
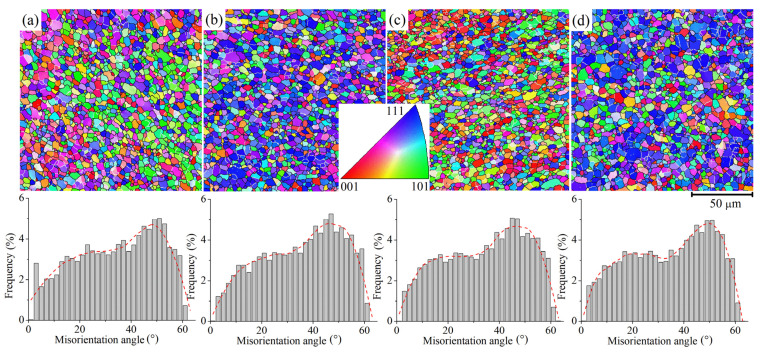
EBSD-IPF maps and misorientation angle distribution histograms corresponding to (**a**) 1HR, (**b**) 2HR, (**c**) CR50, and (**d**) CR80 regimes after 200% superplastic deformation at 480 °C and a 1 × 10^−2^ s^−1^ strain rate.

**Figure 9 materials-16-00445-f009:**
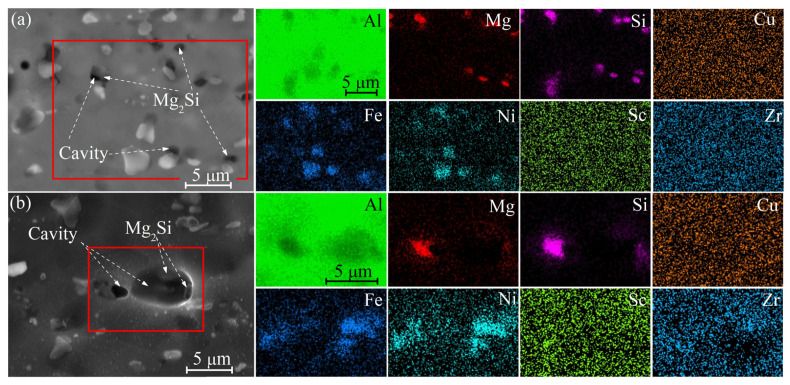
The microstructure and corresponding EDS-maps for samples after the 200% superplastic deformation at 480 °C with a strain rate of 1 × 10^−2^ s^−1^: (**a**) HR2 and (**b**) CR80 processing models.

**Figure 10 materials-16-00445-f010:**
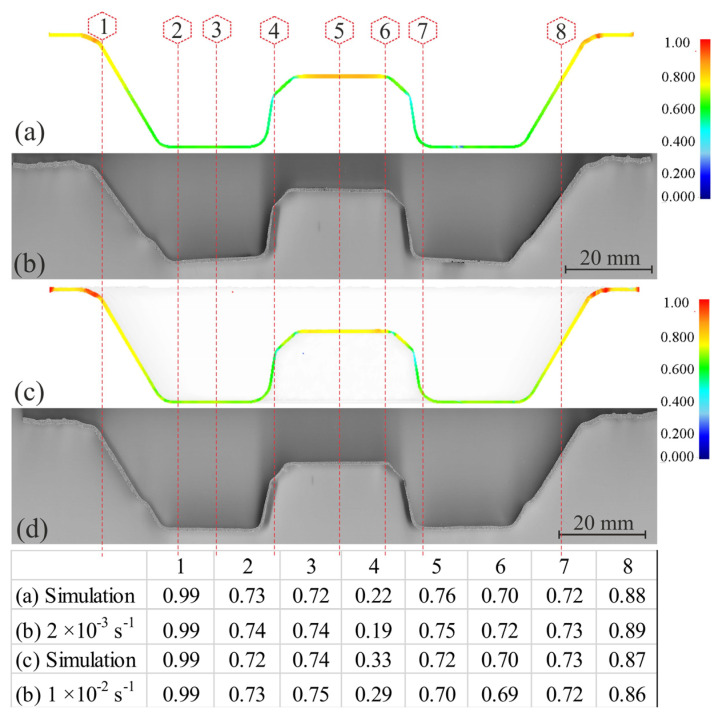
The median cross section and thickness of the part obtained by: (**a**,**c**) SPF and (**b**,**d**) FES by DEFORM 3D at different strain rates, (**a**,**b**) 2 × 10^−3^ s^−1^, and (**c**,**d**) 1 × 10^−2^ s^−1^.

**Figure 11 materials-16-00445-f011:**
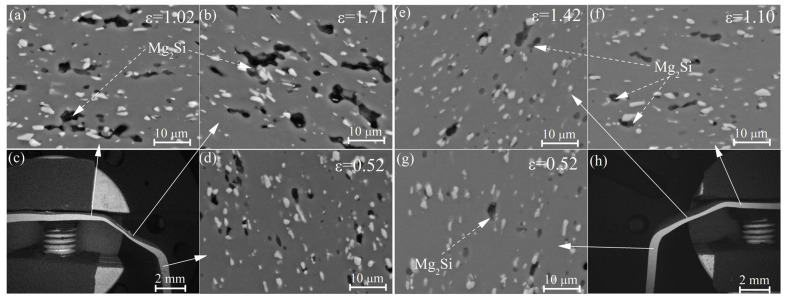
The microstructure in the cross section of the part obtained by SPF with strain rates of (**a**–**d**) 2 × 10^−3^ s^−1^ and (**e**–**h**) 1 × 10^−2^ s^−1^.

**Table 1 materials-16-00445-t001:** Parameters of the thermomechanical treatment regimes.

Regime	Annealing Regime for As-Cast Alloy	Reduction at Hot Rolling (%)	Reduction at Cold Rolling (%)
1HR	350 °C, 8h	95	0
2HR	350 °C, 8h + 480 °C, 3h	95	0
CR50		90	50
CR80		75	80

**Table 2 materials-16-00445-t002:** Microstructure parameters for various regimes of thermomechanical treatment.

Microstructural Parameter	Regimes			
	1HR	2HR	CR50	CR80
Al_9_FeNi particle size (µm)	0.9 ± 0.1	0.9 ± 0.1	0.7 ± 0.1	0.7 ± 0.1
Al_9_FeNi particle aspect ratio	0.8	0.8	0.72	0.72
Mg_2_Si particle size (µm)	0.5 ± 0.1	0.8 ± 0.1	0.8 ± 0.1	0.8 ± 0.1
Mg_2_Si aspect ratio	0.71	0.8	0.8	0.8
Interparticle space	1.4 ± 0.3	1.2 ± 0.1	1.1 ± 0.2	1.1 ± 0.3
Standard deviation for interparticle space	0.9	0.5	0.7	0.9

**Table 3 materials-16-00445-t003:** Elongation-to-failure (%) for the studied alloy treated by different regimes after tensile tests at 480 °C.

Constant Strain Rate (s^−1^)	Treatment Regime			
	1HR	2HR	CR50	CR80
2 × 10^−3^	348 ± 8	447 ± 5	370 ± 10	420 ± 20
5 × 10^−3^	348 ± 12	442 ± 7	380 ± 10	390 ± 25
1 × 10^−2^	353 ± 5	470 ± 5	420 ± 16	450 ± 20

**Table 4 materials-16-00445-t004:** Elongation-to-failure (%) and stress values at a steady stage (true strain of ε = 1) for the 2HR-treated samples at different strain rates and temperatures.

Strain Rate (s^−1^)	Temperature, °C							
	440	460	480	500	440	460	480	500
	Elongation (%)				σ at ε = 1 (MPa)			
2 × 10^−3^	450 ± 8	458 ± 5	447 ± 5	452 ± 8	17	14	13	13
5 × 10^−3^	371 ± 7	498 ± 5	442 ± 7	357 ± 10	23	17	17	17
8 × 10^−3^	376 ± 5	388 ± 5	452 ± 5	375 ± 8	26	20	18	19
1 × 10^−2^	380 ± 7	390 ± 6	470 ± 5	343 ± 10	27	23	21	20
2 × 10^−2^	267 ± 5	267 ± 5	352 ± 7	317 ± 5	32	28	27	27

## Data Availability

Not applicable.

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
