# Peer review of "Microstructure Evolution, Constitutive Modelling, and Superplastic Forming of Experimental 6XXX-Type Alloys Processed with Different Thermomechanical Treatments"

_materials, 2023, doi:10.3390/ma16010445_

Round 1
Reviewer 1 Report
The paper focus on the causes of superplasticity in 6XXX alloys and investigates the role of cavities, second phase particles and the effect of heat treatment on superplasticity. However, there are some minor issues in the paper that need to be clarified by the authors.
The authors have obtained the shape and distribution of the cavities as shown in Figure 11. Line 111 - Line 113 also describes the preparation process of the specimens, but when I made the OM specimens of the aluminium alloy, the polishing was hard. How do you prepared the OM specimens? This should be described in more detail. How do you distinguish cavities and particles in OM and SEM?
2. More detailed information on the experimental setup, test parameters etc. in Fig. 10 should be presented in Section 2.
Author Response
The authors are grateful to the reviewer for the useful comments. The answers are given one by one below. The proper correction was added to the manuscript and marked-up.
Reviewer 1
- The authors have obtained the shape and distribution of the cavities as shown in Figure 11. Line 111 - Line 113 also describes the preparation process of the specimens, but when I made the OM specimens of the aluminium alloy, the polishing was hard. How do you prepared the OM specimens? This should be described in more detail. How do you distinguish cavities and particles in OM and SEM?
The samples for microstructural examination were prepared using a Struers LaboPoll-5 polishing machine via mechanical grinding on SiC papers (grit sizes of 320, 800, 1200, 2400, 4000) and final polishing with an OP-S silica-based colloidal suspension (grain size of 0.04 µm). The grain structure was studied with OM in a polarized light. For this purpose, the pre-polished samples were anodized at a voltage of 18 V in a Barker’s solution for 60 s at a temperature of 2°C below zero. The cavities were studied by SEM in a secondary electron regime. The cavities exhibited dark contrast in the images.
- More detailed information on the experimental setup, test parameters etc. in Fig. 10 should be presented in Section 2.
We thank the reviewer for his valuable comment. Superplastic forming was processed at 480 °C and strain rates of 2×10-3 and 1×10-2 s-1 in a laboratory forming machine with control the Ar gas pressure and temperature of the process using the regime obtained by DEFORM 3D based on the data of the tensile tests. Particularly, the shape and dimensions of the used mold were described in our previous works [1,2]. The used mold shape was designed to have a critical region with different strain rates to evaluate the superplasticity at this region and assess the thickness difference. The particular information on the SPF regime (pressure-time dependence) cannot be provided as a twofold purpose technology
- Mosleh, A.O.; Kotov, A.D.; Mestre-Rinn, P.; Mikhaylovskaya, A.V. Superplastic forming of Ti-4Al-3Mo-1V alloy: flow behavior modelling and finite element simulation. Procedia Manuf. 2019, 37, 239–246, doi:10.1016/j.promfg.2019.12.042.
- Mosleh, A.O.; Mikhaylovskaya, A.V.; Kotov, A.D.; Kwame, J.S. Experimental, modelling and simulation of an approach for optimizing the superplastic forming of Ti-6%Al-4%V titanium alloy. J. Manuf. Process. 2019, 45, 262–272, doi:10.1016/j.jmapro.2019.06.033.

Reviewer 2 Report
In this article, an aluminum alloy has been cast and then subjected to heat treatment and rolling under different conditions. The microstructural properties have been comprehensively investigated. Also, the results are well organized, but the introduction and research method need to be reformed.
The title is too long and can be edited.
Line 94:Use of repeated words “was prepared using a laboratory inductive furnace (Interselt, Saint-Petersburg, Russia) using a graphite-fireclay crucible”
The introduction can be written more clearly and more up-to-date sources can be used. The following relevant and related papers are suggested in this field.
· The influence of post-annealing and ultrasonic vibration on the formability of multilayered Al5052/MgAZ31B composite
· Effects of minor Nd and Er additions on the precipitation evolution and dynamic recrystallization behavior of Mg–6.0Zn–0.5Mn alloy.
· Phase transformations in an ultralight BCC Mg alloy during anisothermal ageing
· Investigation of annealing treatment on the interfacial and mechanical properties of Al5052/Cu multilayered composites subjected to ARB process
· Microstructural origin and control mechanism of the mixed grain structure in Ni-based superalloys
What is the basis of the planning? The reason for choosing heat treatment conditions (temperature and time), ad the amount of applied strain?
Why are the applied strains not the same so that the results can be compared?
How many rolling cycles have the strains been applied?
Separate the modeling section. What is Constitutive modeling used? How are its parameters obtained for this particular alloy? How has its validity been checked?
Author Response
The authors are grateful to the reviewer for the useful comments. The answers are given one by one below. The proper correction was added to the manuscript and marked-up
Reviewer 2
In this article, an aluminum alloy has been cast and then subjected to heat treatment and rolling under different conditions. The microstructural properties have been comprehensively investigated. Also, the results are well organized, but the introduction and research method need to be reformed.
The title is too long and can be edited.
The title is clearly reflected the content of the manuscript and help readers to search the paper in databases.
Line 94: Use of repeated words “was prepared using a laboratory inductive furnace (Interselt, Saint-Petersburg, Russia) using a graphite-fireclay crucible”
Corrected.
The introduction can be written more clearly and more up-to-date sources can be used. The following relevant and related papers are suggested in this field.
- The influence of post-annealing and ultrasonic vibration on the formability of multilayered Al5052/MgAZ31B composite
- Effects of minor Nd and Er additions on the precipitation evolution and dynamic recrystallization behavior of Mg–6.0Zn–0.5Mn alloy.
- Phase transformations in an ultralight BCC Mg alloy during anisothermal ageing
- Investigation of annealing treatment on the interfacial and mechanical properties of Al5052/Cu multilayered composites subjected to ARB process
- Microstructural origin and control mechanism of the mixed grain structure in Ni-based superalloys
We have revised an introduction part and analyzed several recently published papers [1–8].
The paper “Effects of minor Nd and Er additions on the precipitation evolution and dynamic recrystallization behavior of Mg–6.0Zn–0.5Mn alloy” was also added to the reference list due to a similar to our alloy PSN effect observed in the alloys. Unfortunately, the authors find the other recommended papers have no relation with this study.
- Sun, Y.B.; Chen, X.P.; Xie, J.; Wang, C.; An, Y.F.; Liu, Q. High strain rate superplasticity and secondary strain hardening of Al-Mg-Sc-Zr alloy produced by friction stir processing. Mater. Today Commun. 2022, 33, 104217, doi:10.1016/j.mtcomm.2022.104217.
- Lei, G.; Wang, B.; Lu, J.; Wang, C.; Li, Y.; Luo, F. Microstructure, mechanical properties, and corrosion resistance of continuous heating aging 6013 aluminum alloy. J. Mater. Res. Technol. 2022, 18, 370–383, doi:10.1016/j.jmrt.2022.02.101.
- Dumbre, J.; Kairy, S.K.; Anber, E.; Langan, T.; Taheri, M.L.; Dorin, T.; Birbilis, N. Understanding the formation of (Al,Si)3Sc and V-phase (AlSc2Si2) in Al-Si-Sc alloys via ex situ heat treatments and in situ transmission electron microscopy studies. J. Alloys Compd. 2021, 861, 158511, doi:10.1016/j.jallcom.2020.158511.
- Elasheri, A.; Elgallad, E.M.; Parson, N.; Chen, X. ‐Gran. Effect of Si Level on the Evolution of Zr‐Bearing Dispersoids and the Related Hot Deformation and Recrystallization Behaviors in Al–Si–Mg 6xxx Alloys. Adv. Eng. Mater. 2022, 24, 2101606, doi:10.1002/adem.202101606.
- Algendy, A.Y.; Liu, K.; Rometsch, P.; Parson, N.; Chen, X.G. Evolution of discontinuous/continuous Al3(Sc,Zr) precipitation in Al-Mg-Mn 5083 alloy during thermomechanical process and its impact on tensile properties. Mater. Charact. 2022, 192, 112241, doi:10.1016/j.matchar.2022.112241.
- Elasheri, A.; Elgallad, E.M.; Parson, N.; Chen, X.-G. Evolution of Zr-Bearing Dispersoids during Homogenization and Their Effects on Hot Deformation and Recrystallization Resistance in Al-0.8%Mg-1.0%Si Alloy. J. Mater. Eng. Perform. 2021, 30, 7851–7862, doi:10.1007/s11665-021-05917-8.
- Elasheri, A.; Elgallad, E.M.; Parson, N.; Chen, X.-G. Improving the dispersoid distribution and recrystallization resistance of a Zr-containing 6xxx alloy using two-step homogenization. Philos. Mag. 2022, 102, 2345–2361, doi:10.1080/14786435.2022.2103597.
- Elasheri, A.; Elgallad, E.M.; Parson, N.; Chen, X.-G. Nucleation and transformation of Zr-bearing dispersoids in Al–Mg–Si 6xxx alloys. J. Mater. Res. 2022, doi:10.1557/s43578-022-00852-3.
What is the basis of the planning? The reason for choosing heat treatment conditions (temperature and time), ad the amount of applied strain?
The heat treatment conditions were chosen based on our previous result for the alloys with similar composition. It was repeatedly reported that the Sc and Zr bearing alloys require two step homogenization to provide high density of nanoscale dispersoids [9,10]. The first step should be in the range of 300-350 °C whereas the second step accelerate the growth of dispersoids and provide the fragmentation of eutectic originated phases that requires increased temperature of 450-500 °C.
The ratio of cold and hot rolling was taken in the typical range for superplastic Al-based materials.
- Bobruk, E. V.; Dolzhenko, P.D.; Murashkin, M.Y.; Valiev, R.Z.; Enikeev, N.A. The Microstructure and Strength of UFG 6060 Alloy after Superplastic Deformation at a Lower Homologous Temperature. Materials (Basel). 2022, 15, 6983, doi:10.3390/ma15196983.
- Li, Y.; Lu, B.; Yu, W.; Fu, J.; Xu, G.; Wang, Z. Two-stage homogenization of Al-Zn-Mg-Cu-Zr alloy processed by twin-roll casting to improve L12 Al3Zr precipitation, recrystallization resistance, and performance. J. Alloys Compd. 2021, 882, 160789, doi:10.1016/j.jallcom.2021.160789.
Why are the applied strains not the same so that the results can be compared?
The total strain for all thermomechanical regimes was similar. The initial thickness was 20 mm and the final 1 mm, that mean ~ (-3) logarithmic strain. The 1HR and 2HR regimes provided 95 % of hot rolling reduction that mean deformation from 20 mm to 1 mm (-3 logarithmic strain). The CR50 regime provided hot rolling from 20 to 2 mm (90 % or -2.3 logarithmic strain) and then cold rolling from 2 to 1 mm (50 % or -0.7 logarithmic strain). The CR80 regime provided hot rolling from 20 to 5 mm (75 % or -1.4 logarithmic strain) and then cold rolling from 5 to 1 mm (80 % or -1.6 logarithmic strain). This mean that the total reduction was similar for all regimes, and the difference was in the ration of cold and hot rolling, that allows comparing the obtained results.
How many rolling cycles have the strains been applied?
The total strain was accumulated by 10 rolling cycles.
Separate the modeling section. What is Constitutive modeling used? How are its parameters obtained for this particular alloy? How has its validity been checked?
The modelling section was separated under “3.3. Constitutive modeling of the superplastic deformation” paragraph.
The used constitutive model is the Arrhenius constitutive model type using the Simple power equation
ε˙=AF(σ)exp[−Q/(RT)]
Where, F(σ) is a function of stress, F(σ)= σn
The simple power equation was selected in this work due to the obtained low flow stress of the investigated alloy at low strain rates. The description of determining the parameters is described in our previous work [11]. Regarding the validity of the contracture model, it was evaluated by predicting unmodeled data presented in Figure 7a, 440 °C-0.005/s, 460 °C-0.008/s, 480 °C-0.01/s, and 500 °C-0.002/s and the predicted data were compared with the experimental data which approved the good predictability of the constructed model Figure 7c.
- Mosleh, A.O.; Mikhaylovskaya, A. V.; Kotov, A.D.; Kwame, J.S.; Aksenov, S.A. Superplasticity of Ti-6Al-4V Titanium Alloy: Microstructure Evolution and Constitutive Modelling. Materials (Basel). 2019, 12, 1756, doi:10.3390/ma12111756.
